# STYLE DECOMPOSITION AND CONTENT PRESERVATION FOR ARTISTIC STYLE TRANSFER

## ABSTRACT

Artistic style transfer is a crucial task that aims to transfer the artistic style of a style image to a content image, generating a new image with preserved content and a distinct style. With the advancement of image generation methods, significant progress has been made in artistic style transfer. However, the existing methods face two key challenges: i) style ambiguity, due to inadequate definition of style, making it difficult to transfer certain style attributes; ii) content nonrestraint, the lack of effective constraint information causes stylistic features of the content, such as color and texture, to seriously influence content preservation effectiveness. To address this challenges, improving the quality of style transfer while ensuring effective content preservation, we propose SDCP, Style Decomposition and Content Preservation for Artistic Style Transfer, to achieve effective style transfer through style decomposition and content preservation. First, distinguishing from previous work, we propose a style decomposing module that effectively represents style based on three basic attributes (brushstrokes, color, and texture) enabling clear style definition. Second, we design a content preserving module that employs line drawings as constraints to discard style elements while preserving content, utilizing cross-modal alignment to preserving semantic. Finally, all representations are injected into the denoising U-Net through a conditional injection mechanism. Quantitative and qualitative experiments are conducted to demonstrate that SDCP outperforms the current state-of-the-art models.

## 1 INTRODUCTION

Artistic style transfer aims to transfer the artistic style of a style image to a content image and achieve content preservation, generating a new artistic style image, which is important in the field of computer vision and graphics. The key challenge lies in how to achieve effective style feature transfer while maintaining the content unchanged. Traditional methods Zhang et al. (2022; 2024) using generative models Goodfellow et al. (2014); Vaswani (2017) have made a lot of progress. In recent years, breakthroughs in large-scale diffusion models in the field of image generation have inspired more and more style transfer methods Xu et al. (2025); Chung et al. (2024); Gao et al. (2024); Zhang et al. (2023c) to adopt pre-trained Stable Diffusion models Rombach et al. (2022). Although these methods show some promising results, the ambiguity in defining style prevents the effective extraction of its fundamental attributes, thereby hindering the effectiveness of style transfer. Furthermore, they lack effective constraints on content, resulting in insufficient preservation of content details and certain semantic deviations.

First, the results of style transfer lack complete style attributes (textures, colors, and brushstrokes) Qi et al. (2024). Previous methods only perform ambiguous style representation, resulting in inaccurate style expression and omission of some basic style attributes. Specifically, StyleID Chung et al. (2024) treats the style image as two latent spaces (K, V), resulting in imper-

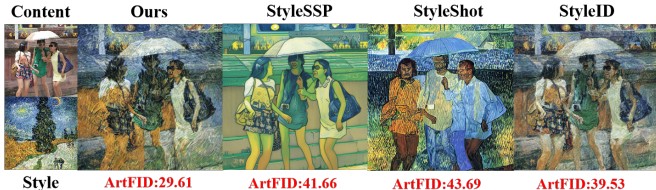

Figure 1: **Style ambiguity**. The results of StyleID, StyleShot, and StyleSSP show that style transfer is less effective. Lower ArtFID indicates better style.

fect style decoupling effects, which are insufficient for constructing style. StyleSSP Xu et al. (2025) employs the whole embedding from the style image encoder's output for style injection, resulting in leakage of the final content and poor content preservation. Styleshot Gao et al. (2024) performs multi-level extraction of style, which effectively mitigates the influence of the content in the style image but also disrupts the fundamental attributes of the style. For example, as shown in Fig. 1, StyleSSP Xu et al. (2025) is ineffective in color transfer, with the result showing a predominantly green tone, which is inconsistent with the multi-tone colors (yellow, blue, white, and green) of the style image. StyleShot Gao et al. (2024) underperforms in texture, which differs significantly from the texture of the style image. StyleID Chung et al. (2024) performs poorly in terms of brushstrokes, which are a characteristic feature of Van Gogh's paintings.

Second, due to the lack of content structure and semantic constraints, many methods face challenges in preserving content structure and maintaining original semantic consistency. StyleID Chung et al. (2024) employs only a simple latent space (Q) as the representation of content without imposing any constraints on it. However, during the reversal process, it is susceptible to the influence of the latent

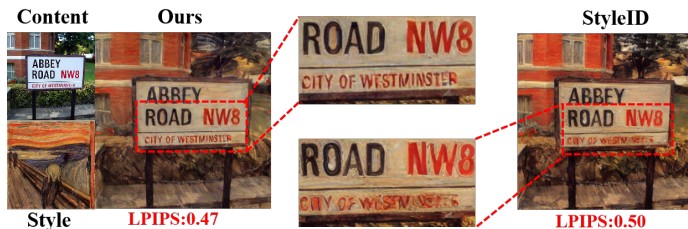

Figure 2: Content nonrestraint. The results of StyleID are deficient in content details. The lower LPIPS Zhang et al. (2018), the better the content is preserved.

spaces from style injection. As shown in Fig. 2, the letters "CITY OF WESTMINSTER" are clearly displayed in the content image, but the image generated by StyleID Chung et al. (2024) is unable to view these words.

To address the two challenges, we design a style decomposing module for style mining and a content preservation module for content details preserving. Firstly, for style mining, we decompose style attribute embeddings from three perspectives: *textures, colors, and brushstrokes*. For colors, we calculate color redundancy to obtain embeddings; for textures, we employ singular value sharing to obtain texture representations specific to grayscale and real images; for brushstrokes, we propose to use quadratic Bézier curves to approximate brushstroke information in artworks simulating painting brushstrokes to obtain important brushstroke features (length, width, and maximum curvature). Secondly, for content preserving, we propose to use line drawings as structure constraint for content, as line drawings lack stylistic interference and preserve content details. To accomplish that, we design a line drawing generation method that extracts detailed content information from the content image while discarding its stylistic information. Additionally, we enhance semantic constraints by introducing content text and performing cross-modal alignment using Q-former Li et al. (2023b). Finally, to ensure that the denoising U-Net Ronneberger et al. (2015) effectively adapts to the injection of style and content embeddings, we propose a conditional injection mechanism based on the cross-attention mechanism, using a denoising U-Net as the base model, to effectively combine style and content embeddings for generating new images. The main contributions of this paper are as follows:

- We propose a novel framework called SDCP, as shown in Fig. 3, which enhances style transfer quality while content preservation, leading to more accurate style representation and cleaner content extraction.

- We propose a style decomposing module that captures the key features of style through three style properties: brushstroke, color, and texture. This solves problem of style ambiguity and enables effective artistic style transfer.

- We design a content preserving module that effectively preserves content through structural and semantic constraints, overcoming the content nonrestraint issue.

- In regard to automated and manual evaluation metrics, extensive empirical evaluations demonstrate both qualitatively and quantitatively that our method significantly improves the quality of artistic style transfer.

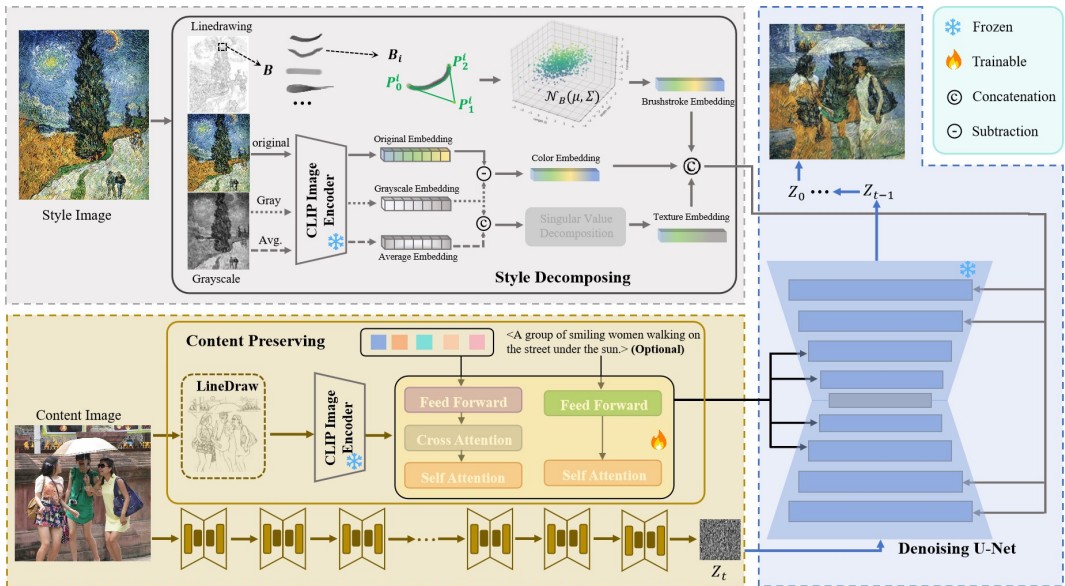

Figure 3: The overview of our proposed SDCP. It contains three parts, (a) content preserving module, which carves the content details and discards the original style of the content image, (b) style decomposing module, which reconstructs image style from three perspectives, namely colors, textures, and brushstrokes, (c) condition injecting mechanism, which implements combination of style and content conditions.

## 2 RELATED WORK

**GAN-Based Methods.** A large number of studies Zhu et al. (2017); Liu et al. (2018); Karras (2019); Li et al. (2020); Xu et al. (2021); Zhang et al. (2022) have broadened the application of GANs Goodfellow et al. (2014) while addressing specific challenges in style transfer. Significant progress Zhang et al. (2022) has also been made in enhancing style transfer results through contrastive learning methods. A notable example is CAST Zhang et al. (2022), which uses contrastive learning to directly encode style codes from image features. This approach improves style distribution learning, leading to enhanced stylization quality and consistency. Although GAN-based methods were earliest started to be applied to the field of style transfer, the unstable training and the necessity of supervised labeling for assistance are still large challenges.

**Transformer-Based Methods.** StyleFormer Wu et al. (2021) effectively solves the common content representation bias problem in traditional neural style transfer methods by integrating style library generation. StyTr$^2$ Deng et al. (2022) utilizes two independent encoders to generate domain-specific content and style sequences. In addition, S2WAT Zhang et al. (2024) achieves a finer and more efficient style transfer with a hierarchical vision attention transformer. Overall, many transformer-based Zhang et al. (2023a); Li et al. (2023c); Liu et al. (2024) methods are excellent for style transfer. Still, because transformer was designed to associate contextual semantic information, it makes final result carry content semantic information about style images.

**Diffusion-Based Methods.** Diffusion Rombach et al. (2022) has attracted a great deal of attention since its appearance in the field of image generation. The same phenomenon has occurred in the field of artistic style transfer. InST Zhang et al. (2023c) derives directly from a single simplified style conversion process and improves the quality and efficiency of the generated images without the requirement of complex textual descriptions. DEADiff Qi et al. (2024) achieves high-fidelity, controllable image style transfer by explicitly decoupling and reassigning content and style features in the latent space of the diffusion process. Its core lies in the "dual-path cross-attention" mechanism, which can precisely inject arbitrary style details while maintaining the content structure. StyleID Chung et al. (2024) is a training-free method for fine-tuning large-scale diffusion models, thus simplifying the process and improving efficiency. In general, InST Zhang et al. (2023c) and StyleID Chung et al. (2024) are not able to maintain their content efficiently due to multiple noisy interferences (style images and text prompts) during the reversal process.

## 3 PROPOSED METHOD

### 3.1 PRELIMINARIES

**Diffusion Model**. LDM Rombach et al. (2022) is a perceptual compression model based on DDPM Ho et al. (2020) to access an efficient and low-dimensional latent space to reduce computational cost. The fundamental idea is to define a Markov chain that gradually transforms a data sample into pure noise, and then train a neural network to approximate the reverse dynamics in order to recover the original data distribution Ho et al. (2020). LDM consists of two main processes: diffusion process (forward process) and denoising process (reverse process). To be specific, given data sampled from $q(x_0)$, the forward diffusion process is defined as a sequence of Gaussian transitions:

$$q(x_t \mid x_{t-1}) = \mathcal{N}\big(x_t; \sqrt{1-\beta_t}x_{t-1},\, \beta_t I\big), \tag{1}$$

where $\{\beta_t\}_{t=1}^T$ is a variance schedule controlling the noise intensity at each step. As $t \to T$, the distribution of $x_T$ approaches an isotropic Gaussian, which makes sampling from the terminal distribution straightforward. Importantly, this process admits a closed-form expression that directly relates $x_t$ to the initial clean sample $x_0$:

$$q(x_t \mid x_0) = \mathcal{N}\big(x_t; \sqrt{\bar{\alpha}_t}x_0,\, (1-\bar{\alpha}_t)I\big), \tag{2}$$

with $\alpha_t = 1 - \beta_t$ and $\bar{\alpha}_t = \prod_{s=1}^t \alpha_s$. This forward process progressively injects noise into the data until all structure and features are lost, which can be closely approximated by $\mathcal{N}(0, I)$.

The generative reverse process is parameterized as another Gaussian transition:

$$p_\theta(x_{t-1} \mid x_t) = \mathcal{N}\big(x_{t-1}; \mu_\theta(x_t, t),\, \Sigma_\theta(x_t, t)\big), \tag{3}$$

where $\mu_\theta$ and $\Sigma_\theta$ are predicted by a neural network (UNet or Transformer) with learnable parameters $\theta$. Since the true reverse posterior is intractable, the model is trained to approximate it by minimizing a variational bound. A practical simplification, introduced in Ho et al. (2020), shows that the objective reduces to predicting the added noise $\epsilon$ with a mean squared error loss:

$$L(\theta) = \mathbb{E}_{x_0, t, \epsilon \sim \mathcal{N}(0, I)} \big[\, \|\epsilon - \epsilon_\theta(x_t, t)\|^2 \,\big] \tag{4}$$

### 3.2 STYLE DECOMPOSING MODULE

Inspired by vector line art Mo et al. (2021), we believe that brushstroke characteristics in artistic style images are an important attribute of style features, which were often overlooked in previous work. Specifically, different artists exhibit distinct brushstroke characteristics. As shown in Fig. 4, we see the differences in the brushstrokes of these five painters. Picasso: Points are concentrated in shorter, thinner, and lower-curvature regions, reflecting his geometric and minimalist lines; Seurat: High-curvature, short brushstrokes are prominent, consistent with his pointillist style; Monet: Brushstrokes are relatively wide with moderate curvature, reflecting his fluid and soft Impressionist style; Cezanne: Distribution is relatively compact, showcasing his structured and repetitive brushstroke patterns; Van Gogh: Brushstrokes are the largest, thickest, and highest in curvature, aligning with his wild lines.

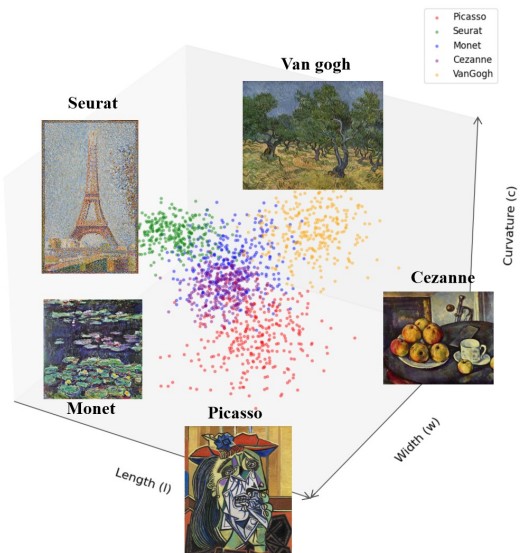

Figure 4: The figure shows the distribution of strokes extracted from the works of five painters (Picasso, Seurat, Monet, Cezanne, and Van Gogh) projected into three-dimensional space based on three attributes: stroke length (l), width (w), and curvature (c).

**Brushstrokes Extraction**. We elaborate the detailed process of stroke feature extraction. First, LineDraw is used to generate the line drawing of the style image. Then, the linedrawing vectorisation method Mo et al. (2021) is used to obtain all strokes $B = \{(B_i, w_i)\}_i^N$ in the image. Specifically, each stroke $B_i$ is approximated using a quadratic Bézier curve, defined by three points $P_0^i$, $P_1^i$, and $P_2^i$, where $w_i$ represents the width of the stroke. $B_i = (1-\tau)^2 P_1^i + 2\tau(1-\tau)P_1^i + \tau^2 P_2^i$, where $\tau \in [0, 1]$, $P_0$ and $P_2$ are the start and end points, $P_1$ is the control point, and $N$ is the total number of strokes. As shown in Fig. 4, we aim to construct the stroke distribution for each style image, which requires three basic elements: length, maximum curvature, and width.

According to Graveson Gravesen (1997) approximate arc length $L_{\text{approx}}$ calculation method, the curve length depends on the chord length $L_{\text{chord}}$ (the distance between the start and end points) and the polygon length $L_{\text{chord}}$ (the distance between each point and the control point)

$$L_{\text{approx}} = \frac{L_{\text{poly}} + 2L_{\text{chord}}}{3}, \tag{5}$$

where $L_{\text{chord}} = \|P_2 - P_0\|$, $L_{\text{poly}} = \|P_1 - P_0\| + \|P_2 - P_1\|$.

Similarly, the arc length $l_i$ of $B_i$ can be represented as

$$l_i = \frac{\|P_1^i - P_0^i\| + \|P_2^i - P_1^i\| + 2\|P_2^i - P_0^i\|}{3}. \tag{6}$$

The curvature of $B_i$ is defined as: $\kappa(\tau) = \frac{\|B_i'(\tau) \times B_i''(\tau)\|}{\|B_i''(\tau)\|^3}$, where $B_i'(\tau) = 2[(1-\tau)(P_1^i - P_0^i) + \tau(P_2^i - P_1^i)]$, $B_i''(\tau) = 2(P_2^i - 2P_1^i + P_0^i)$. $B_i'(\tau)$ varies with changes in $\tau$. When $\|B_i''(\tau)\| = 0$, $B_i''(\tau)$ reaches its minimum value, and thus $\kappa(\tau)$ reaches its maximum value. Therefore, the maximum curvature of $B_i$ is

$$c_i = \frac{(P_0^i - P_1^i) \cdot (P_2^i - 2P_1^i + P_0^i)}{\|P_0^i - 2P_1^i + P_2^i\|^2}. \tag{7}$$

Each stroke $B_i$ can be represented by an attribute vector $v_i = (l_i, w_i, c_i)$, and all strokes in a style image can be represented by a $V \in \mathbb{R}^{3 \times N}$. Therefore, we can model the stroke style as a trivariate normal distribution $v \sim \mathcal{N}_V(\mu, \Sigma)$, where $\mu$ and $\Sigma$ are the mean and variance of the matrix $V$, respectively. Finally, the stroke style embedding $E_{brs}$ is obtained by sampling from $\mathcal{N}(\mu, \Sigma)$.

**Colors Extraction**. Inspired by the color additivity in the text prompt space Brack et al. (2023), we believe that color additivity in the image space holds true as well Ye et al. (2023). The essence of color feature extraction lies in how to decouple the content and texture semantics of style images, isolating only the color features. We adopt an approach to achieve color decoupling. First, we convert the style image to grayscale $(GS)$, such that $GS(I_S)$ retains the content and texture semantics while discarding all colors distributions. Then, we employ the CLIP image encoder to encode the original image $I_S$ and $GS(I_S)$. Finally, the color attributes can be represented as,

$$E_{clr} = Enc(I_S) - Enc(GS(I_S)). \tag{8}$$

**Textures Extraction**. For texture extraction, we also adopt grayscale processing to eliminate the influence of color. However, it causes interference from grey tones when extracting textures. To overcome this limitation, we concatenate embeddings of grayscale image $Enc(GS(I_S))$ and average grayscale image $Enc(AVG(GS(I_S)))$, the latter carrying overall greyscale tone. The initial texture embedding is represented as,

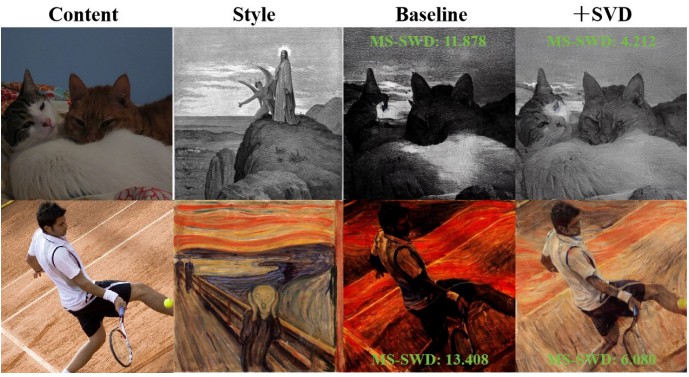

Figure 5: Examples of SVD. Smaller MS-SWD He et al. (2024) indicates smaller color distances between the style and the result.

$$E_{tex}^* = Concat(Enc(GS(I_S)), Enc(AVG(GS(I_S)))). \tag{9}$$

Inspired by previous work Li et al. (2024), we observe that the singular values of $E_{tex}^*$ represent the shared information between the two grayscale images, i.e., the grayscale tones. Next, we use singular value decomposition to capture the texture information, $E_{tex}^* = U\Sigma V^T$, where $\Sigma = diag(\sigma_0, \sigma_1, \cdots, \sigma_{nj})$. To suppress the influence of grey tone on texture features, we constrained all singular values, i.e., $\hat{\sigma} = \sigma e^{-\sigma}$. Then, we update $E_{tex} = U\hat{\Sigma}V^T$, where $\hat{\Sigma} = diag(\hat{\sigma}_0, \hat{\sigma}_1, \cdots, \hat{\sigma}_{nj})$. Some examples are shown in the Fig.5. Ultimately, the style embedding of style image $I_S$ is defined as:

$$E_S = Concat(E_{brs}, E_{tex}, E_{clr}). \tag{10}$$

## 3.3 CONTENT PRESERVING MODULE

In order to achieve content preservation and better acquire delicate content details of content images, we design two content constraint paradigms.

**Structure Constraint**. Unlike previous work, we use line drawings of the content rather than whole content images as the content structure. The structure of the content image can be effectively represented by line drawings, which minimize stylistic interference. To extract clear and detailed line drawings of content, we design a line drawing generation method called LineDraw. In detail, as shown in Fig. 6, LineDraw accomplishes transformation of an image from a real domain to a line drawing through

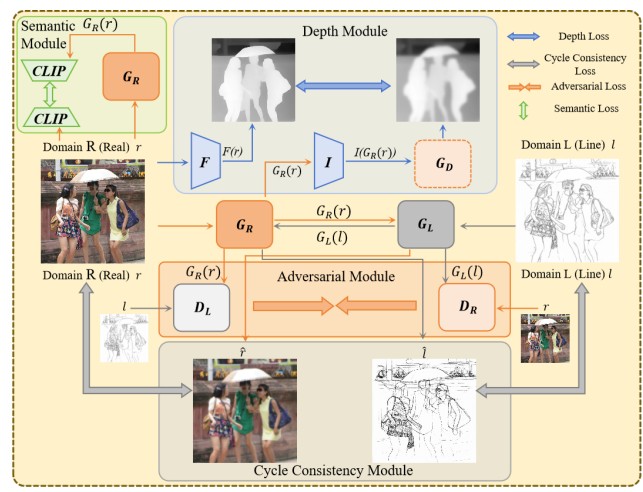

Figure 6: The workflow of our proposed LineDraw.

a generator $\mathcal{G}_R$, on the contrary, $\mathcal{G}_L$ generates a corresponding real image. LineDraw mainly contains depth module, cycle consistency module, semantic module, and adversarial module.

Depth module aims at obtaining more depth information of a real image $r$, i.e., the content image $I_C$. We recognize that depth maps often represent shapes as well as contours of individual entities in an image well. Therefore, we design a depth map generator $\mathcal{G}_D$ and a depth loss constraint $L_{depth}$ such that line drawings generated by $\mathcal{G}_R$ carry depth information. Since, many of existing datasets Lin et al. (2014); Tan et al. (2018) lack corresponding depth maps, we follow the previous model Miangoleh et al. (2021) as depth map model $F$. Depth map $\mathcal{F}(r)$ generated by $F$ is pseudo ground truth map for only training purposes. In addition, due to the domain gap between line drawings and depth maps, we did not directly use line drawings from $\mathcal{G}_R$ as input to $\mathcal{G}_D$ directly, but first extracted line drawing features using $I$ Szegedy et al. (2016) before obtaining depth maps from $\mathcal{G}_D$. This is due to the fact that features within earlier layers are more beneficial for transfer learning Kornblith et al. (2020). Our depth constraint objective is Depth Loss:

$$L_{dep} = \|\mathcal{G}_D(I(\mathcal{G}_R(r))) - \mathcal{F}(r)\|_2^2. \tag{11}$$

Cycle consistency module is designed to balance distribution between different domains of generated images. Image translation cycle should be able to bring images back to original images Zhu et al. (2017), i.e., $r \rightarrow \mathcal{G}_R(r) \rightarrow \mathcal{L}(\mathcal{G}_R(r)) \rightarrow \hat{r} \approx r$. Consequently, our goal for cycle consistency constraint is Cycle Consistency Loss:

$$L_{cyc} = \|\mathcal{G}_L(\mathcal{G}_R(r)) - r\|_2^2 + \|\mathcal{G}_R(\mathcal{G}_L(l)) - l\|_2^2. \tag{12}$$

Adversarial module is designed to train a maximal-minimal game for generator $\mathcal{G}$ and discriminator $\mathcal{D}$ under adversarial objective Goodfellow et al. (2014), and our adversarial constraint objective is Adversarial Loss:

$$L_{adv} = \mathbb{E}_{r \sim R}[\mathcal{D}_R(r)^2] + \mathbb{E}_{l \sim L}[(1 - \mathcal{D}_R(\mathcal{G}_L(l)))^2] + \mathbb{E}_{l \sim L}[\mathcal{D}_L(l)^2] + \mathbb{E}_{r \sim R}[(1 - \mathcal{D}_L(\mathcal{G}_R(r)))^2]. \tag{13}$$

Semantic module aims to maintain semantic consistency between line drawings and real images leveraging CLIP Radford et al. (2021) embeddings. We employ a pre-trained visual coder $Enc$ Radford et al. (2021) to encode semantic information of real images and line drawings separately. Thus, our semantic constraint goal is Semantic Loss:

$$L_{sem} = \|Enc(\mathcal{G}_R(r)) - Enc(r)\|_2^2. \tag{14}$$

Ultimate loss for LineDraw is specified as follows:

$$L = \lambda_{dep}L_{dep} + \lambda_{cyc}L_{cyc} + \lambda_{adv}L_{adv} + \lambda_{sem}L_{sem}. \tag{15}$$

As a summary, given a content image $I_C$, it is directly encoded as a latent spatial representation $Z_t$ by a pre-trained diffusion model $SD$ on the one hand, and on the other hand, line drawing $LD(I_C)$ is passed through the CLIP image encoder $Enc$, yielding the content embedding $E_{cnt}$.

**Semantic Constraint**. Inspired by DEADiff Qi et al. (2024) and BLIP-Diffusion Li et al. (2023a), we aim to obtain a more complete content-centric representation from the line drawing. Therefore, we introduce a Q-former Li et al. (2023b) mechanism that enhances the content-semantic representation of line drawings through cross-modal interaction alignment. Its input includes a learnable query variable, captions (text describing the content image $I_C$) corresponding to the content dataset (i.e., MSCOCO), and the content embedding $E_{cnt}$. Finally, its output is a query embedding vector $E_C$ aligned with the text.

### 3.4 Condition Injecting Mechanism

In order to support style embedding and content embedding as conditions for the denoising U-Net, we propose a conditional joint mechanism based on cross-attention, inspired by the IP-Adapter Ye et al. (2023).

$$Q = ZW^Q, \tag{16}$$

$$K = Concat((E_C W_C^K), (E_S W_S^K)), \tag{17}$$

$$V = Concat((E_C W_C^V), (E_S W_S^V)), \tag{18}$$

$$Z^{new} = Softmax(\frac{QK^T}{\sqrt{d_K}})V. \tag{19}$$

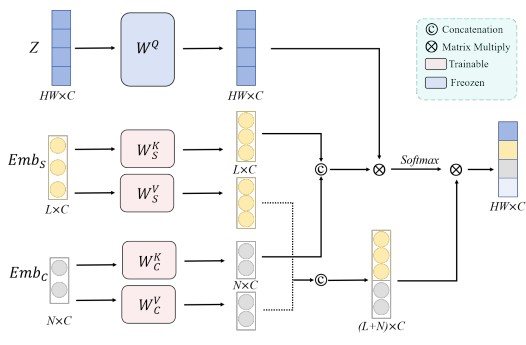

Figure 7: Condition injecting mechanism.

As shown in Fig. 7, it consists of four trainable linear layers $W_C^K, W_S^K, W_C^V, W_S^V$ to combine $E_C$ and $E_S$. Subsequently, the attention mechanism is utilized to form new latent space variables $Z^{new}$. Unlike previous works, cross-attention is no longer used to align cross-modal features but to combine different image embeddings within the same modality.

## 4 Experiments

### 4.1 Implement Details

**Experimental settings.** In the inference stage, we use a pre-trained Stable Diffusion model Rombach et al. (2022) (version 1.5), with time step $T$=50. All experiments are implemented in Pytorch on an NVIDIA RTX 4090 GPU. We set the weighting trade-off parameters of loss functions as follows: $\lambda_{dep} = 10, \lambda_{cyc} = 0.1, \lambda_{adv} = 1, \lambda_{sem} = 10$. For specific experiment details on our method, please refer to the supplementary materials.

**Dataset.** We utilize WikiArt Phillips & Mackintosh (2011) as style images and MSCOCO Lin et al. (2014) as content images. We randomly sample 40 style images from WikiArt and 20 content images from MSCOCO.

**Evaluation Metrics.** We use three metrics (i.e., FID, LPIPS, and ArtFID) to demonstrate the style transfer performance of our model. Frechet Inception Distance (FID) Heusel et al. (2017) is used to measure the overall similarity between a style image and a stylized content image, with smaller values representing fewer semantic features of the style image that the stylized content image possesses. Learned Perceptual Image Patch Similarity (LPIPS) Zhang et al. (2018) is used to estimate

the average perceptual distance between a content image and a stylized content image. We adopt a metric that exclusively evaluates the overall style transfer performance, ArtFID Wright & Ommer (2022). Overall, for artistic style transfer, FID and ArtFID are used to measure the similarity between the stylized result and the style image (style fidelity), LPIPS is used to measure similarity between the stylized result and the content image (content fidelity).

## 4.2 QUANTITATIVE COMPARISONS

**Automatic Metric Comparisons.** In general, for artistic style transfer, FID and ArtFID are used to measure the similarity between the stylized result and the style image (style fidelity), while LPIPS is used to measure the similarity between the stylized result and the content image (content fidelity). Our method achieves excellent results on all three metrics. However, it is worth noting that in Table 1, CAP-VST has the highest LPIPS value. As shown in Fig. 9, the images it generates have a low

| Models | Backbone | ArtFID↓ | FID↓ | LPIPS↓ |
|--------|----------|---------|------|--------|
| Ours | Diffusion | **29.6137** | **19.1558** | 0.4693 |
| StyleSSP | Diffusion | 41.6647 | 27.2383 | 0.4754 |
| StyleShot | Diffusion | 43.6397 | 24.4324 | 0.7159 |
| StyleID | Diffusion | 39.5329 | 25.2136 | 0.5081 |
| MambaST | Mamba | 35.5372 | 20.8257 | 0.6282 |
| S2WAT | Transformer | 41.7495 | 26.0384 | 0.5440 |
| InST | Diffusion | 44.1601 | 24.6718 | 0.7202 |
| CAP-VST | GAN | 51.5054 | 39.5442 | **0.2703** |
| CAST | GAN | 40.4869 | 25.1618 | 0.5475 |
| StyTR$^2$ | Transformer | 39.0501 | 24.0271 | 0.5603 |

Table 1: Quantitative comparisons of SOTA models. Bold font indicates optimal scores.

degree of stylization but a high degree of content preservation (i.e., high LPIPS), but the lowest scores on the style metrics (ArtFID, FID). This demonstrates that the metrics we selected are consistent with the visual effects.

**User Preference Study.** Since the evaluation of style transfer performance involves a considerable degree of subjective judgment, we conducted a user preference study (4 questions, each with 3 options) to quantitatively evaluate our method, with 30 volunteers participating in the study. Specifically, we invite each participant to select the method name corresponding to the best stylized result in each of the four dimensions: content preservation, colors transfer, textures transfer, and brushstrokes transfer (method names are anonymised). The results of the user preference study are shown in Figure 8. For style, our method outperforms other methods in terms of color, brushstrokes, and texture. Regarding content preserving, our method achieves the same level as the current state-of-the-art method StyleSSP.

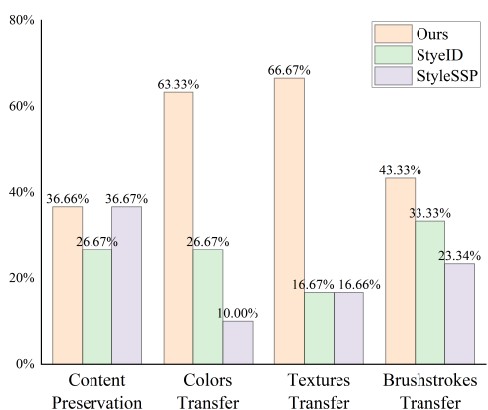

Figure 8: User preference study. We report the overall preference score comparing our method to selected alternatives across three transfer tasks.

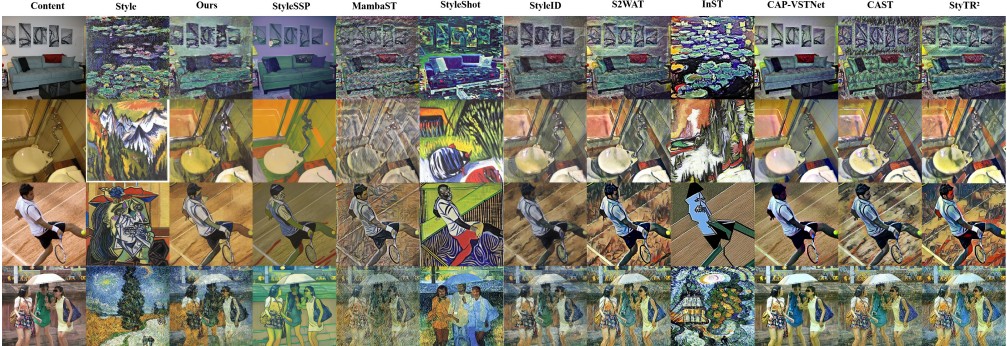

Figure 9: Qualitative comparisons of SOTA models.

### 4.3 QUALITATIVE COMPARISONS

These compared models are classified into four categories according to backbone, Diffusion-based models: StyleSSP Xu et al. (2025), StyleID Chung et al. (2024), StyleShot Gao et al. (2024), InST Zhang et al. (2023c), Transformer-based models: S2WAT Zhang et al. (2024), StyTR$^2$ Deng et al. (2022), GAN-based models: CAP-VSTNet Wen et al. (2023), CAST Zhang et al. (2022), Mamba-based model: MambaST Botti et al. (2024). More results are provided in the appendix.

Qualitative comparisons are presented in Fig. 9. Although some diffusion-based baseline methods capture partial style information, such as StyleSSP retaining color style and StyleID identifying simple textures, results from the last row show that these methods struggle to extract brushstroke information within the style image. Other baseline methods acquire richer style information but lack effective content preservation, e.g., StyleShot and InST. Compared to GAN-based methods (CAST and CAP-VST), we observe that their generated output retains content more effectively, but it also carries more style information related to the content. Regarding Transformer-based methods (S2WAT and StyTR$^2$), we note that they can't effectively transfer style to the result.

### 4.4 ABLATION STUDY

In this section, we present ablation studies results for two aspects of our method: i) the impact of style decomposing, ii) the impact of content preservation.

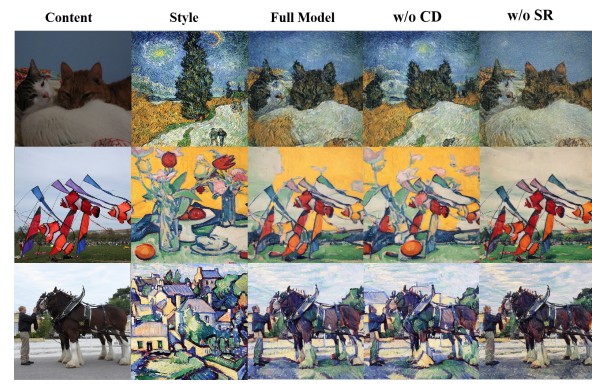

**Style decomposing.** As shown in Fig. 12, without the style decomposing module, synthesized results differ significantly from the style image in terms of brushstrokes, colors, and textures. At the same time, ArtFID and FID have increased in Table 2, which also verifies the fact that the style decomposition module is beneficial for style transfer.

Figure 10: Visualization results of ablation study.

**Content preservation.** As shown in Fig. 12, when the content preservation module is removed (i.e., without content embedding), the model results exhibit severe content loss and style-content hallucinations in terms of content preservation. For example, the results in the first row show that the black-and-white cat face on the left is no longer clear, and the ears disappear. Additionally, its high LPIPS value further validates the decline in content preservation.

Furthermore, to better analyze the impact of each embedding on the stylized results within the style decomposition module, we conducted ablation studies for the three attributes: Colors, Textures, and Brushstrokes. Quantitative results show that all three attribute embeddings have a significant impact on the overall style. More ablation studies are available in appendix.

| Model | ArtFID↓ | FID↓ | LPIPS↓ |
|---|---|---|---|
| Full Model | **29.6137** | **19.1558** | **0.4693** |
| w/o CD | 35.3061 | 19.3646 | 0.7337 |
| w/o SR | 36.9621 | 24.5107 | 0.5080 |
| w/o Colors | 36.1546 | 22.2108 | 0.5576 |
| w/o Textures | 34.1693 | 21.5332 | 0.5164 |
| w/o Brushstrokes | 35.5861 | 20.5885 | 0.6483 |

Table 2: Various quantitative results of ablation experiments.

### 5 CONCLUSION

In this paper, we propose SDCP, a new framework for artistic style transfer. We discover two key challenges in style transfer methods: style ambiguity and content nonrestraint. To address these challenges, we introduce two key components: (1) style decomposition to achieve clearer style representation, (2) content preservation to effectively preserve content details. Experimental results demonstrate that SDCP effectively reduces the loss of original content while showing excellent style transfer performance. Comparative experiments show that our method outperforms relevant baselines both qualitatively and quantitatively. Future work can explore more complex artistic styles to further enhance style transfer across diverse scenarios and improve lightweight solutions.

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

## A    LLM USAGE STATEMENT

A large language model (LLM) was used solely for language polishing. The LLM did not contribute to the research design, development of methods, data analysis, or the formulation of conclusions. All scientific claims, results, and interpretations in this paper are entirely the responsibility of the authors.

## B    DETAILED ANALYSIS OF TEXTURES EXTRACTION

Inspired by soft-weighted regularization Li et al. (2024), we use SVD to extract texture style from $E_{tex}^*$. We construct an embedding matrix $E_{tex}^* = [Enc(GS(I_S)), Enc(AVG(GS(I_S)))]$ from the image encoder in CLIP, where $Enc(AVG(GS(I_S)))$ is actually a grayscale image encoding (derived from the style grayscale map). We utilize SVD,

$$E_{tex}^* = U\Sigma V^T \tag{20}$$

where $\Sigma = diag(\sigma_0, \sigma_1, \cdots, \sigma_{nj})$, and here $\sigma_{nj}$ represents the value of the n-th row and j-th column in the $\Sigma$ matrix.

Then, to suppress grayscale information, we introduced soft weighting regularization for each singular value.

$$\hat{\sigma} = \sigma e^{-\sigma} \tag{21}$$

To visually demonstrate the SVD process, we illustrate the specific dimensional changes of the singular value decomposition at the top of Fig. 11. Additionally, in Fig. 11, we visualize the changes in the $\Sigma$ matrix before and after soft-weighted regularization. Finally, the bottom of Fig. 11 shows the style results before and after soft weight regularization. We find that Equation 2 is effective in weakening the grayscale effect.

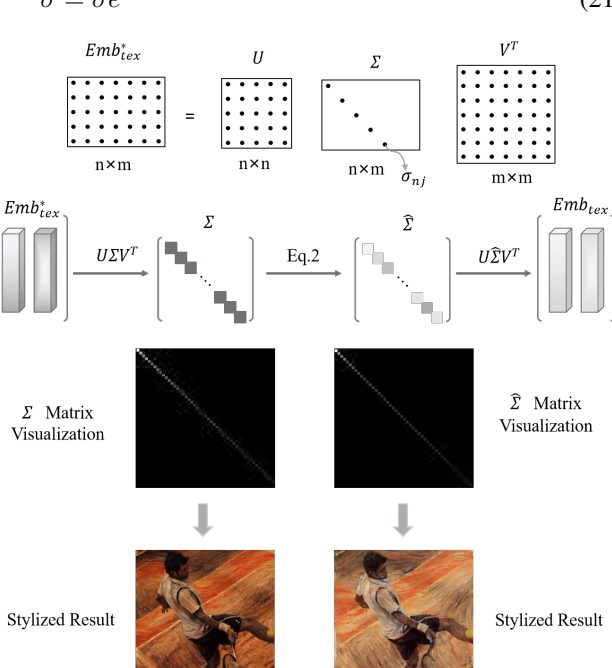

Figure 11: The workflow of singular value decomposition and soft weight regularization.

## C    DETAILED ANALYSIS OF ABLATION STUDY

In this section, we supplement the ablation studies presented in the paper. Specifically, we conduct both qualitative and quantitative ablation studies on linedraw, Q-former, and three attribute embeddings.

As shown in Fig. 12 and Tab. 3, in terms of colors, when without color embedding, the cat's color aligns with the color of the content image. When lacking brushstroke information, Van Gogh's unique style (short, dense brushstrokes) cannot be effectively transferred. In terms of texture, when texture embedding is absent, the brushstrokes are not rendered perfectly.

| Content | Style | Full Model | w/o Linedrawing | w/o Q-former | w/o Textures | w/o Colors | w/o Brushstrokes |

Figure 12: Visualization results of ablation experiments.

Additionally, as shown in the Tab. 3, these three style attributes play a crucial role in the final style transfer. In terms of content, the combination of Q-former and line drawings significantly improves the overall content retention.

| Model | ArtFID↓ | FID↓ | LPIPS↓ |
|---|---|---|---|
| Full Model | **29.6137** | **19.1558** | **0.4693** |
| w/o LineDraw | 35.4829 | 20.1107 | 0.6808 |
| w/o Q-former | 35.2667 | 20.0021 | 0.6792 |
| w/o Colors | 36.1546 | 22.2108 | 0.5576 |
| w/o Textures | 34.1693 | 21.5332 | 0.5164 |
| w/o Brushstrokes | 35.5861 | 20.5885 | 0.6483 |

Table 3: Various quantitative results of ablation experiments.

# D DETAILS OF LINEDRAW METHOD

In this section, we provide more details about LineDraw, mainly containing the network architecture of the method. In addition, we show experimental results for the method.

## D.1 DATASETS

We randomly selected 10,000 images from MSCOCO Lin et al. (2014) as the real image training dataset. We adopt 14,914 sketch images from the Anime Sketch Colorization Pair [1] training dataset as our line drawing training dataset.

## D.2 ABLATION STUDY

The ablation study is conducted targeting the losses of the proposed depth module and semantic module as shown in Fig. 14. From left to right are the content images (from MSCOCO Lin et al. (2014)), the results of the variant model without $L_{depth}$, the results of the variant model without $L_{sem}$, and the results of the full model. By comparing the $2^{nd}$ and lastest columns, it can be seen that $L_{depth}$ makes the general outline of the objects in the image clearer, such as the outline of the shape of a fish. By comparing the $3^{th}$ and lastest column, it can be seen that $L_{sem}$ makes the details of the objects in the image richer, e.g., the eyes of the fish are portrayed.

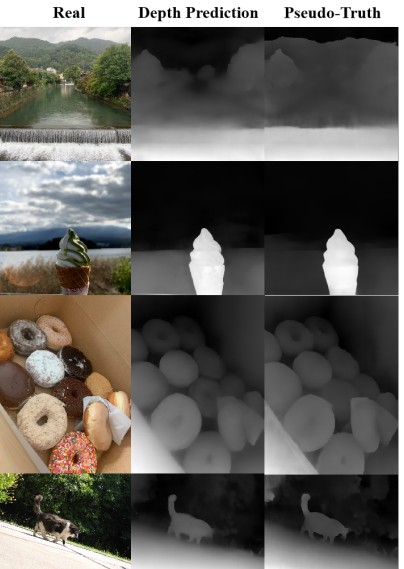

Figure 13: The depth prediction results by $G_D$ are compared to pseudo-ground truth results.

[1]https://www.kaggle.com/datasets/ktaebum/anime-sketch-colorization-pair

In addition, we show the generation results of the proposed depth map of $G_D$ and compare it with the pseudo-ground truth results (by $F$ Miangoleh et al. (2021)), as shown in Fig. 13. The experiment proves that our trained depth map generator $G_D$, although not able to fully achieve the effect of $F$, is able to basically portray the contour of the objective entity in the image, which plays a well-assisting role in the overall line drawing map generation.

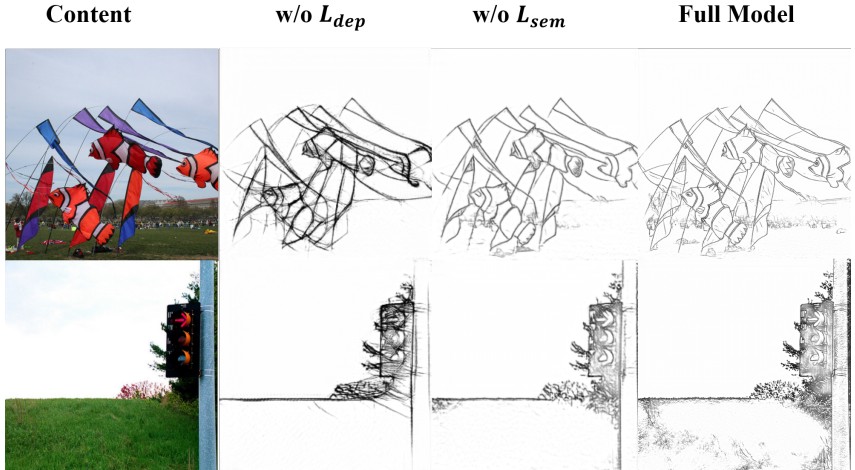

Figure 14: Various variants results of the ablation study on LineDraw.

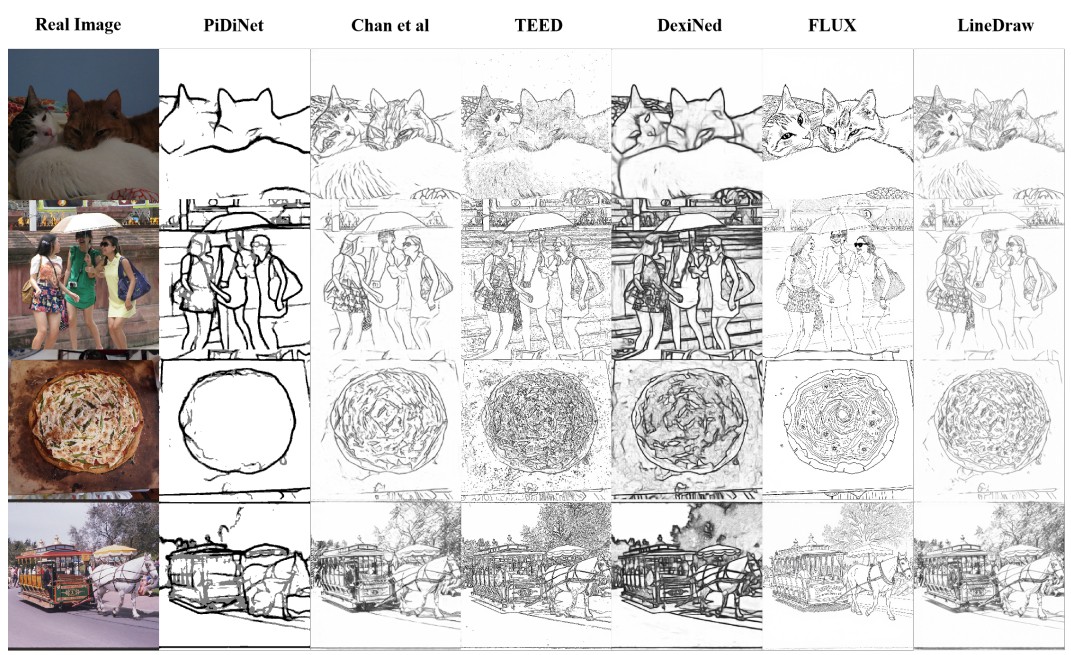

Figure 15: Qualitative comparison of line drawing generation results.

### D.3 QUALITATIVE COMPARISON

We compare the line drawing generation results of our proposed LineDraw and baselines (FLUX Labs et al. (2025), ControlNet Zhang et al. (2023b), TEED Soria et al. (2023a), DexiNed Soria et al. (2023b), Chan et al Chan et al. (2022), and PiDiNet Su et al. (2021)). As shown in Fig. 15, although Chan et al. can display some contours, there is insufficient detail. In addition, TEED captures better detail, but it also introduces more noise data (i.e., noise points). Unlike these methods, our method neither introduces noise data that affects the semantic expression of line drawings nor compromises reliable detail information.

### D.4 NETWORK ARCHITECTURES

The generator network $G_R$ use the encoder-decoder architecture with 3 residual blocks He et al. (2016). The discriminator networks $D_R$ and $D_L$ are according to PatchGAN Isola et al. (2017). We show the architecture details of $G_R$ as shown in Table 4 and the specific architecture of $G_D$ as shown in Table 5.

| Layer Type | Padding | Kernel Size | Stride | Normalization | Activation | Input, Output |
|---|---|---|---|---|---|---|
| Conv2d | 0 | $7 \times 7$ | 1 | InstanceNorm | ReLU | 3,64 |
| Conv2d | 1 | $3 \times 3$ | 2 | InstanceNorm | ReLU | 64,128 |
| Conv2d | 1 | $3 \times 3$ | 2 | InstanceNorm | ReLU | 128,256 |
| ResidualBlock | 1 | $3 \times 3$ | 1 | InstanceNorm | ReLU | 256,256 |
| ResidualBlock | 1 | $3 \times 3$ | 1 | InstanceNorm | ReLU | 256,256 |
| ResidualBlock | 1 | $3 \times 3$ | 1 | InstanceNorm | ReLU | 256,256 |
| ConvTranspose2d | 1 | $3 \times 3$ | 2 | InstanceNorm | ReLU | 256,128 |
| ConvTranspose2d | 1 | $3 \times 3$ | 2 | InstanceNorm | ReLU | 128,64 |
| Conv2d | 1 | $7 \times 7$ | 1 | InstanceNorm | Sigmod | 64,1 |

Table 4: Network Architecture of $G_R$.

| Layer Type | Padding | Kernel Size | Stride | Normalization | Activation | Input,Output |
|---|---|---|---|---|---|---|
| Conv2d | 4 | $7 \times 7$ | 1 | BatchNorm | ReLU | 768,512 |
| ConvTranspose2d | 0 | $4 \times 4$ | 2 | BatchNorm | ReLU | 512,256 |
| ResidualBlock | 1 | $3 \times 3$ | 1 | BatchNorm | ReLU | 256,256 |
| ResidualBlock | 1 | $3 \times 3$ | 1 | BatchNorm | ReLU | 256,256 |
| ResidualBlock | 1 | $3 \times 3$ | 1 | BatchNorm | ReLU | 256,256 |
| ResidualBlock | 1 | $3 \times 3$ | 1 | BatchNorm | ReLU | 256,256 |
| ResidualBlock | 1 | $3 \times 3$ | 1 | BatchNorm | ReLU | 256,256 |
| ResidualBlock | 1 | $3 \times 3$ | 1 | BatchNorm | ReLU | 256,256 |
| ResidualBlock | 1 | $3 \times 3$ | 1 | BatchNorm | ReLU | 256,256 |
| ResidualBlock | 1 | $3 \times 3$ | 1 | BatchNorm | ReLU | 256,256 |
| ResidualBlock | 1 | $3 \times 3$ | 1 | BatchNorm | ReLU | 256,256 |
| ConvTranspose2d | 1 | $3 \times 3$ | 2 | BatchNorm | ReLU | 256,128 |
| ConvTranspose2d | 1 | $3 \times 3$ | 2 | BatchNorm | ReLU | 128,64 |
| ConvTranspose2d | 1 | $3 \times 3$ | 2 | BatchNorm | ReLU | 64,64 |
| Conv2d | 3 | $7 \times 7$ | 1 | BatchNorm | ReLU | 64,3 |

Table 5: Network Architecture of $G_D$.

## E EXPERIMENTAL DETAILS

### E.1 IMPLEMENTATION DETAILS

During the training phase, only the Q-former Li et al. (2023b) and four linear layers ($W_C^K, W_S^K, W_C^V, W_S^V$) need to be updated. All experiments are implemented using NVIDIA RTX 4090 GPUs on PyTorch. The model consists of 16 cross-attention layers, which are numbered from 1 to 16 in order from input to output. Layers 5 to 9 are defined as coarse layers for content embedding, while the remaining layers are considered fine layers for style embedding. The image encoder for CLIP Radford et al. (2021) uses the ViT-L/14 version, and the number of learnable query tokens

in the Q-former is 16. Additionally, to enable the Q-former to converge quickly, we utilize the pre-trained model from BLIP-Diffusion Li et al. (2023a). We use AdamW Loshchilov et al. (2017) as the optimizer with a learning rate of 1 e-4 and train for 1,000 iterations.

For training the Q-former, we use all image-text pairs from the MSCOCO dataset Lin et al. (2014), but since the image-text correspondence in this dataset is one-to-many, it is not conducive to cross-modal semantic alignment. Therefore, for each image, we only used one text caption as the unique image-text pair for training.

## E.2 ADDITIONAL EXPERIMENTAL RESULTS

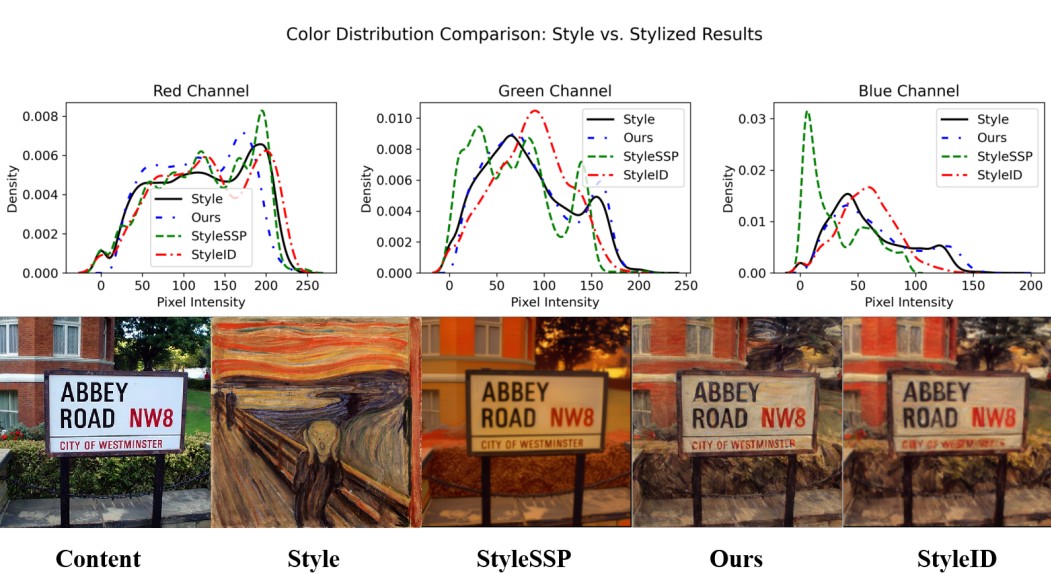

Figure 16: Comparison of color distribution in style transfer examples.

**Color transfer comparison.** We supplement with an example of color transfer in the image, comparing the stylized results with the color distribution curve of the style image, as shown in Fig. 16. Considering the red, green, and blue color channels, we conclude that the color distribution curve of our results is more similar to the style image.

**More experimental results.** We have presented more stylized image results. Due to the large sample size, we are unable to display all 800 stylized images, but the 40 style images are all presented. These stylized images are primarily derived from renowned artists such as Berthold Morisot, Edvard Munch, Ernst Ludwig Kirchner, Jackson Pollock, Monet, Nicholas Roerich, Pablo Picasso, Paul Cézanne, Paul Gauguin, Samuel Peploe, Vincent van Gogh, Wassily Kandinsky, etc.

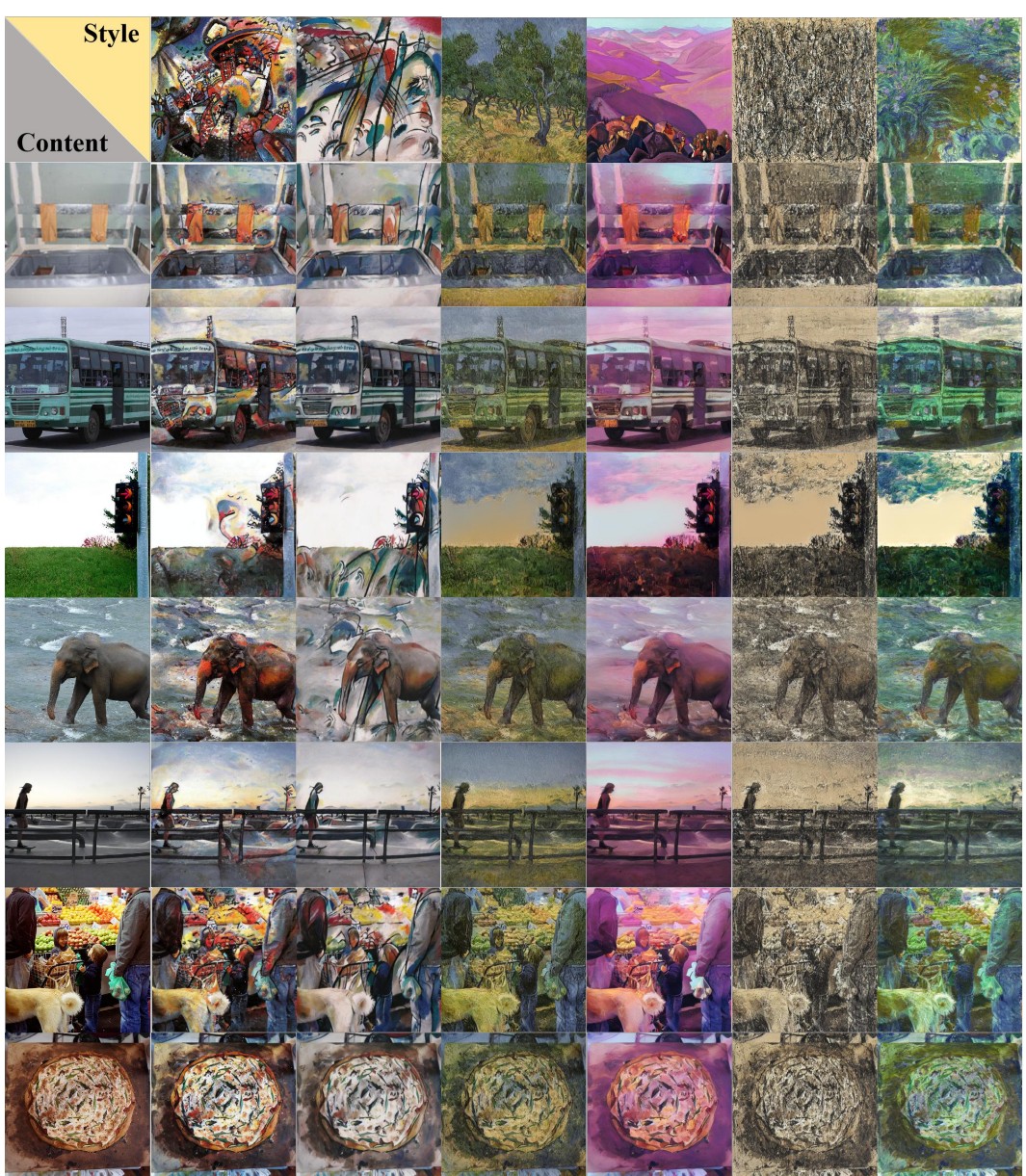

Figure 17: More experimental results of different styles.

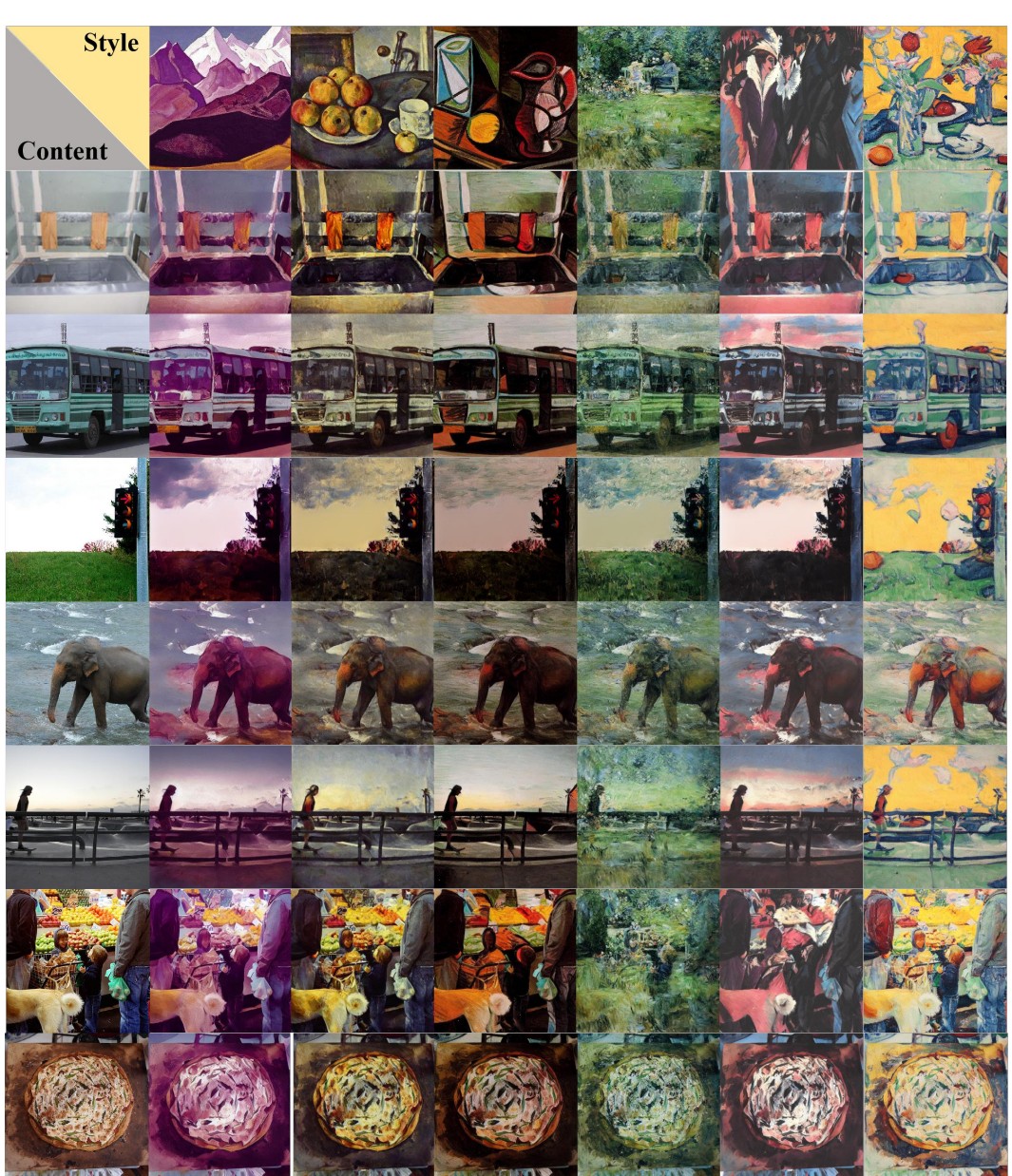

Figure 18: More experimental results of different styles.

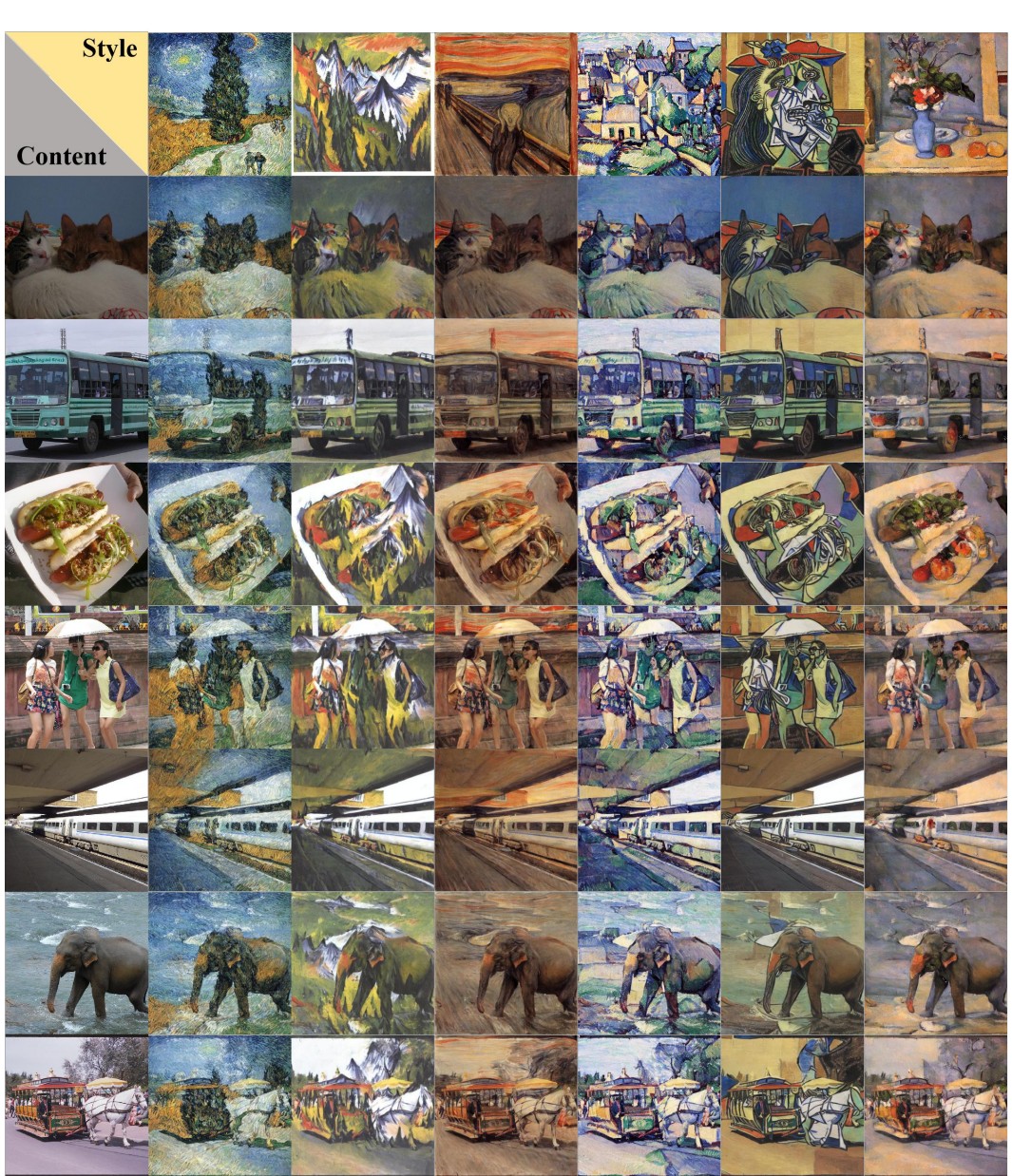

Figure 19: More experimental results of different styles.

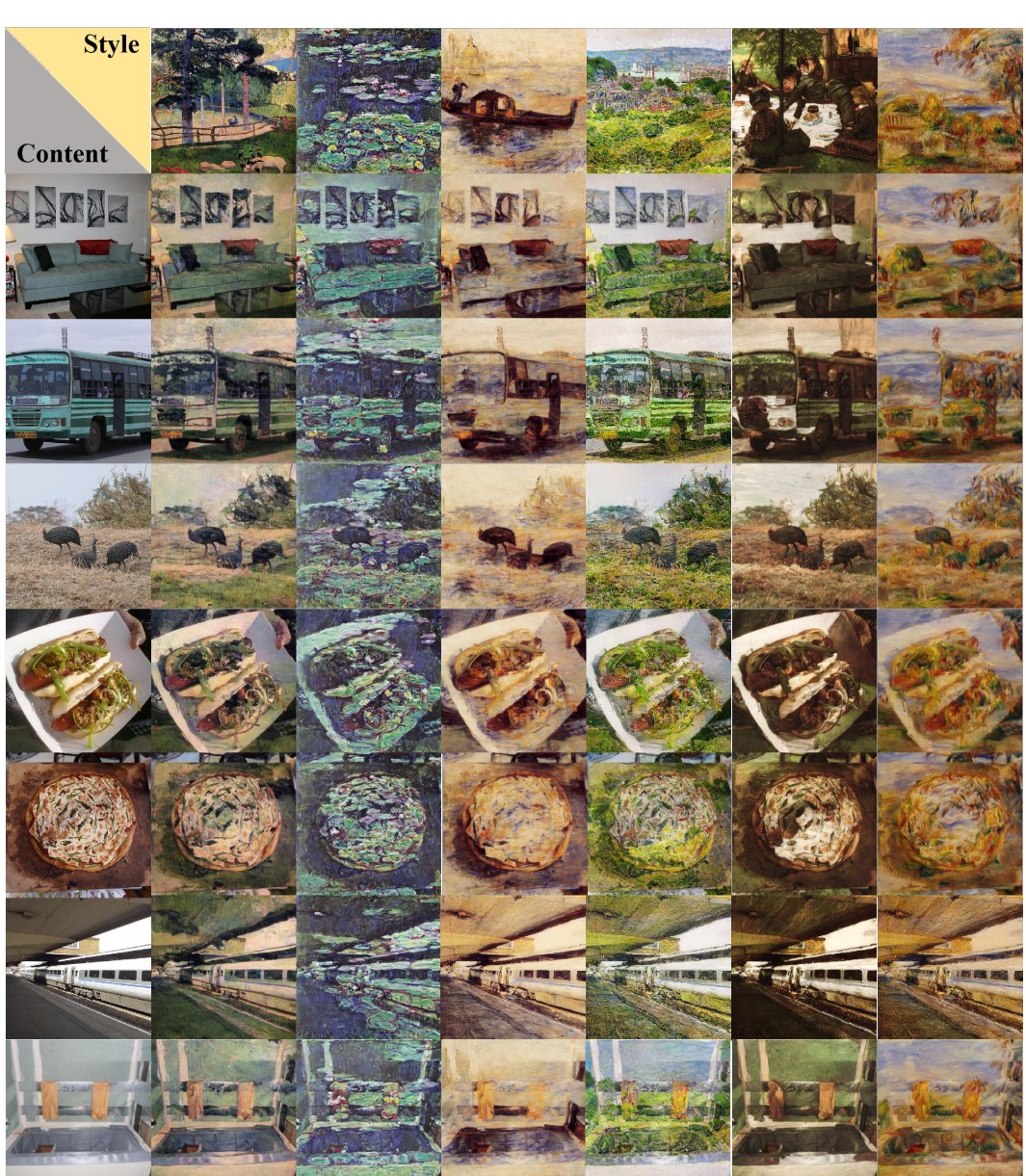

Figure 20: More experimental results of different styles.

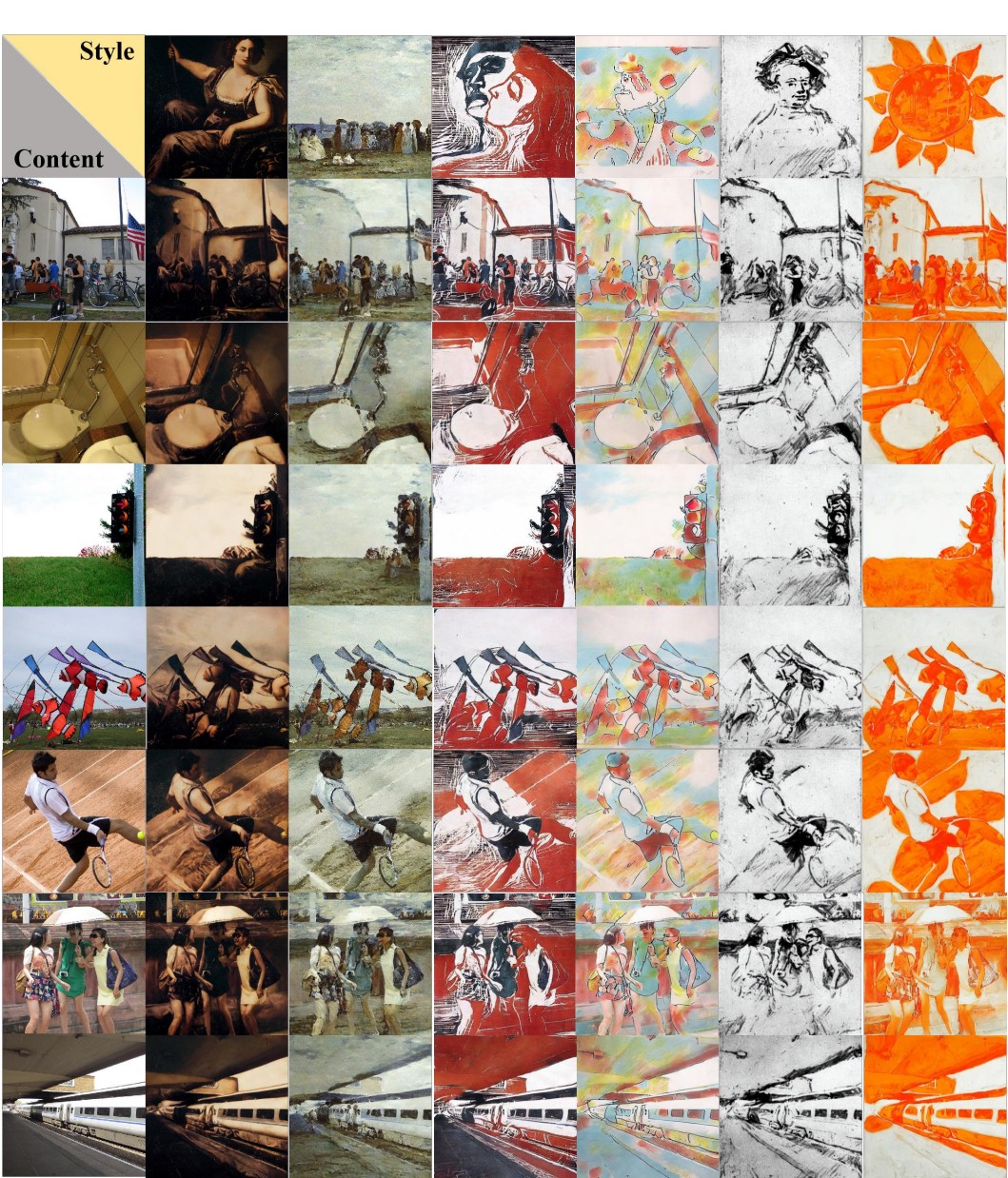

Figure 21: More experimental results of different styles.

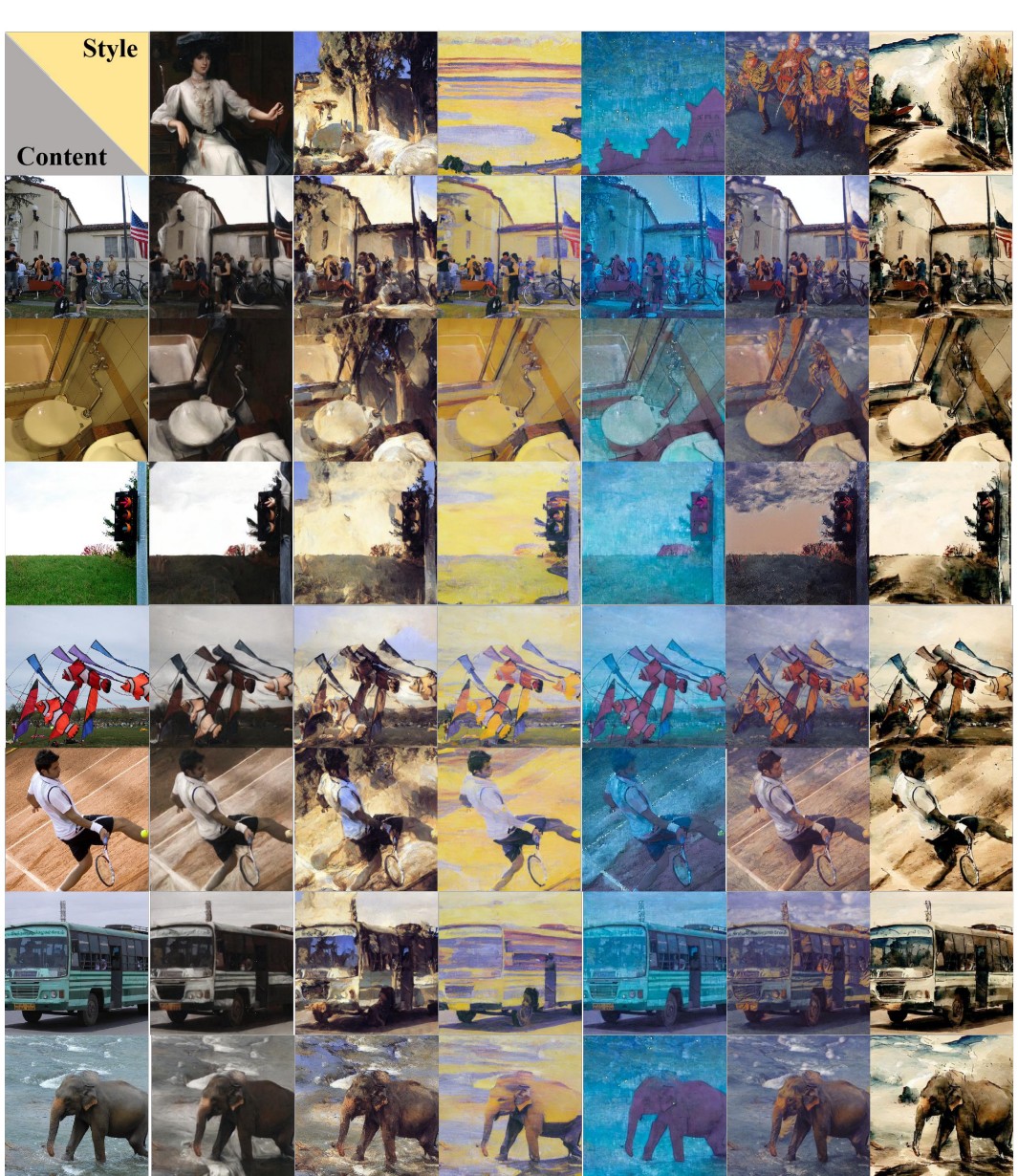

Figure 22: More experimental results of different styles.

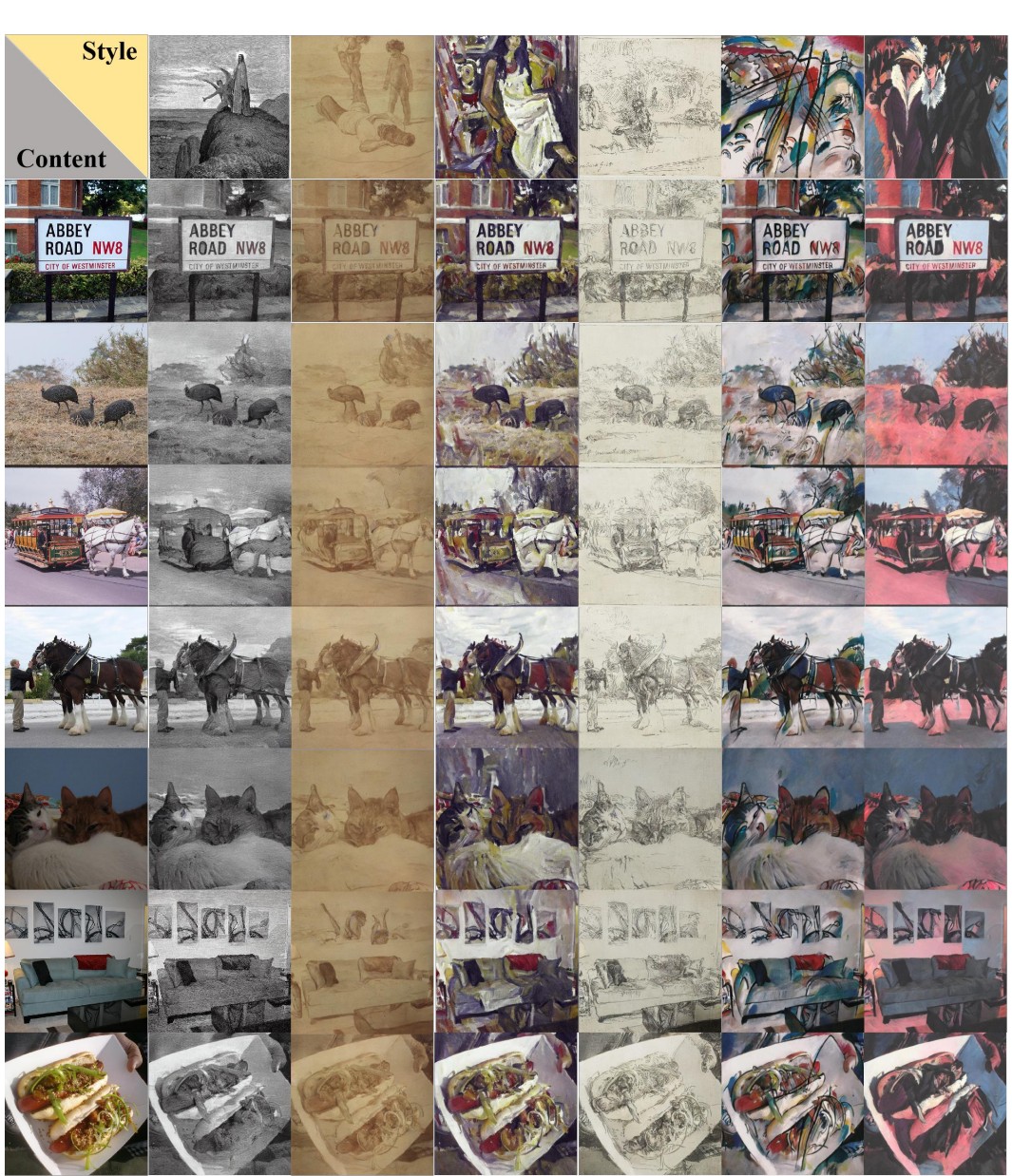

Figure 23: More experimental results of different styles.

