# OpenReview forum: "Style Decomposition and  Content Preservation for Artistic Style Transfer"
_ICLR.cc/2026/Conference — ICLR 2026 Conference Withdrawn Submission_

### Official Review · Reviewer_suBr · 2025-10-17

**Soundness:** 2
**Presentation:** 2
**Contribution:** 2
**Rating:** 4
**Confidence:** 4

**Summary:**

This paper aims to address two key challenges in existing style transfer approaches: style ambiguity and content non-restraint. To this end, the authors propose two core modules: first, a style decomposition module that effectively represents style using three attributes—brushstrokes, color, and texture; second, a content preservation module that leverages line drawings as constraints to filter out style elements while retaining content, with cross-modal alignment employed to preserve semantics. These two representations are then injected into a denoising U-Net via a conditional injection mechanism.

**Strengths:**

+ Using white-box definitions to decompose style is interesting and encouraging.

+ Comprehensive experiments and ablation studies.

**Weaknesses:**

- The proposed method is quite complex, with numerous detailed operations. Notably, the definitions of style components are largely inherited from existing literature, yet there is a lack of clear evidence to substantiate whether these operations are optimal choices. This concern is further underscored by the ablation results in Figure 12, where removing certain representations (e.g., Line drawing, Q-former, textures) appears to exert little impact on the results.

- The authors propose a LineDraw method based on CycleGAN for extracting content line drawings, but the rationale for adopting this relatively traditional approach remains unclear. What advantages does this method offer in comparison to current SOTA line drawing extraction techniques? Additionally, the motivation for using line drawings to enhance content preservation requires further elaboration. Would substituting line drawings with Canny edge maps or grayscale images yield comparable effects?

**Questions:**

Please see weaknesses above.

---

### Official Review · Reviewer_6Kfx · 2025-10-23

**Soundness:** 2
**Presentation:** 3
**Contribution:** 2
**Rating:** 4
**Confidence:** 4

**Summary:**

This paper proposes a new style transfer framework called SDCP to improve the quality of style transfer while ensuring effective content preservation. The proposed method consists of two key components: a style decomposing module that captures the key features of style through three style properties (brushstroke, color, and texture), and a content preserving module that effectively preserves content through structural and semantic constraints.

**Strengths:**

1. This paper aims to solve the problems of style ambiguity and content nonrestraint, which are both challenging and meaningful.
2. This paper proposes representing style based on three basic attributes (brushstrokes, color, and texture), which could offer a new perspective for style transfer.
3. Extensive experiments are conducted to evaluate the performance of the proposed method.

**Weaknesses:**

1. Currently, there are many methods that can generate line drawings from a given image. Why not directly use existing methods? What advantages does the newly designed method, LineDraw, have compared to the current methods?

2. This paper proposes extracting style based on three basic attributes: brushstrokes, color, and texture. However, the results in Figure 12 show that the impact of 'w/o brushstrokes,' 'w/o color,' and 'w/o texture' on the final outcome is not significant. Does this imply that the proposed strategy may not be very effective?

3. The qualitative comparison examples with other methods provided in Figure 9 are too few (only 4 examples), and it seems that the method proposed in this paper does not exhibit significantly superior performance. It is hard to find one example that the proposed approach is significantly better.

4. In the ablation experiment shown in Figure 10, the difference between using and not using content preservation is not significant.

5. What is the time efficiency of different methods? Some evaluations regarding this could be conducted.

**Questions:**

Please see **Weaknesses**.

Others:

1. The experiments in the paper are based on Stable Diffusion 1.5, which is no longer considered state-of-the-art in the community. In addition to newer U-Net architecture models like SDXL 1.0, I am also more interested in whether the proposed method can be applied to models based on the DiT architecture, such as Stable Diffusion 3 and FLUX.1.

2. The effectiveness of the loss functions proposed in Section 3.3 should be validated through ablation studies.

---

### Official Review · Reviewer_RyEq · 2025-10-25

**Soundness:** 3
**Presentation:** 2
**Contribution:** 2
**Rating:** 4
**Confidence:** 5

**Summary:**

The paper proposes SDCP , a framework for artistic style transfer. The main idea is to achieve more effective and interpretable style transfer by separately modeling style and content. The authors introduce a style decomposition module that represents artistic style in terms of three fundamental attributes, namely brushstrokes, color, and texture, which enables clearer style definition and control. In addition, they design a content preservation module that uses line drawings as structural constraints to remove stylistic artifacts while maintaining semantic and spatial consistency through cross-modal alignment. Overall, the method aims to achieve a good balance between stylistic expression and faithful content preservation in artistic style transfer.

**Strengths:**

1、The paper is overall easy to follow.

**Weaknesses:**

1、The authors appear to devote considerable effort to the design of the content preserving module for extracting a line drawing from an image, but this seems unnecessary given that there are already numerous existing methods for extracting line drawings from images.

2、Although I understand that the use of line drawings is intended to inject less content information and encourage the model to generate high-quality images, it also inevitably leads to the loss of fine content details.

3、For U-Net-based architectures, the design appears somewhat outdated. It is recommended that the authors explore more recent approaches based on DiT (Diffusion Transformer) structures.

4、As the paper emphasizes style decomposition and content preservation, it would be more convincing if the evaluation focused on metrics that directly reflect these two aspects, such as style consistency and content preservation, and demonstrated clear improvements over existing methods.

5、There have already been a number of studies exploring style transfer based on DiT architectures, which seems to be clearly lacking in this paper, such as Omnistyle.

**Questions:**

See Weaknesses.

---

### Official Review · Reviewer_VHvW · 2025-11-01

**Soundness:** 2
**Presentation:** 2
**Contribution:** 2
**Rating:** 4
**Confidence:** 4

**Summary:**

This paper proposes SDCP (Style Decomposition and Content Preservation), a diffusion-based framework for artistic style transfer. The authors argue that prior diffusion-based style transfer methods suffer from two main issues:
(1) Style ambiguity — unclear definition of “style” leading to incomplete transfer, and
(2) Content non-restraint — lack of constraints that preserve structural and semantic content.

**Strengths:**

The idea of separating style into brushstrokes, color, and texture is conceptually interesting and aligns with artistic analysis theory.

The paper is well-organized and provides many illustrations and examples (e.g., brushstroke distribution across artists, qualitative comparisons, and ablation visualizations).

**Weaknesses:**

Only 40 style images and 20 content images are used — far too small to support quantitative claims.
The improvements on FID/ArtFID are modest and within possible noise margins.
User study design is unclear (only 30 participants, limited task description).

The proposed framework still relies on Stable Diffusion v1.5, which has become outdated compared to modern DiT-based architectures (e.g., Flux, UViT, or PixArt). Since DiT variants offer better efficiency and spatial fidelity, it is unclear why the authors did not adopt or compare against a transformer-based diffusion backbone.

The system contains too many hand-designed modules (style decomposition, line drawing, Q-former alignment, conditional injection, etc.), making it extremely difficult to isolate which component truly contributes to the improvement. The design feels more like a pipeline of heuristics rather than a coherent, principled model.

The assumption that all styles can be decomposed into brushstrokes, color, and texture is restrictive. Many modern or abstract art styles (e.g., minimalism, photography-based stylization) do not contain clear brushstrokes or texture cues, raising doubts about the generality of this representation.

The paper only compares against older diffusion methods such as StyleID and StyleSSP, but omits newer and stronger baselines like DiffStyler, Paint-by-Example, or other training-free diffusion editing techniques. Without these comparisons, it’s difficult to judge real progress.

The motivation for standalone style transfer is becoming weaker with the rise of general-purpose image editing models (e.g., Qwen-Image-Edit), which can also take an image prompt as a “style input.” The paper does not discuss how SDCP fits into this new paradigm or whether its advantages persist when compared to these modern, prompt-based editing frameworks.

**Questions:**

Have the authors verified whether the claimed improvements still hold if the same design is implemented on a DiT-based architecture such as Flux, PixArt?

Given the large number of modules (e.g., style decomposition, LineDraw, Q-former alignment, conditional injection), how can the reader assess which part contributes most to performance?

Have you conducted any module-level ablation or sensitivity analysis beyond simple removal experiments to isolate the actual effect of each component?

The decomposition assumes that all styles can be described by brushstroke, color, and texture. How does this approach generalize to styles that lack explicit brushstrokes or textures (e.g., minimalist, digital, or photographic styles)?

Would the model fail or degrade gracefully when such attributes are not present?

---

### Note · Authors · 2025-11-14

I have read and agree with the venue's withdrawal policy on behalf of myself and my co-authors.